# Mechanism and Function of Circular RNA in Regulating Solid Tumor Radiosensitivity

**DOI:** 10.3390/ijms231810444

**Published:** 2022-09-09

**Authors:** Junchao Huang, Huihui Sun, Zike Chen, Yingjie Shao, Wendong Gu

**Affiliations:** 1Department of Radiation Oncology, The Third Affiliated Hospital of Soochow University, Changzhou 213003, China; 2Departement of Medicine, Nantong University, Nantong 226019, China

**Keywords:** radiosensitivity, circRNA, tumor, radiotherapy

## Abstract

Radiotherapy is an important tool in the treatment of malignant tumors, and exploring how to make radiotherapy more effective is a new way to break through the current bottleneck in the development of radiation oncology. Circular RNAs (circRNAs) are a special class of endogenous non-coding RNAs. Numerous studies have shown that circRNAs have shown great potential in regulating the biological functions of tumors, including proliferation, migration, invasion, and treatment resistance, and that differences in their expression levels are closely related to the clinical prognosis of tumor patients. This review systematically compares the mechanisms of circRNAs in the process of tumor development and radiosensitivity and provides insight into the clinical translation of circRNAs in radiotherapy.

## 1. Introduction

Circular RNAs (circRNAs) were first discovered in pathogens. As early as 1976, Sanger et al. described viroids containing “single-stranded and covalently closed circRNA molecules” [1]. However, such covalently closed circRNAs were not identified and confirmed in human genes until 1993. However, only a small number of circRNAs were found at that time, so they were only considered abnormal splicing by-products with little functional potential [2,3,4,5]. Moreover, many studies only classify circRNAs as scrambled exons, products of exon rearrangements, or just non-linear mRNAs [6,7,8]. Recently, advances in RNA-seq techniques have led to an explosion in the research on circRNAs. An increasing number of studies have also found that circRNAs are implicated in a variety of biological processes [4,5,9].

circRNAs can be mainly divided into four categories: exonic circRNAs (ecircRNAs) [10], intronic circRNAs [11], exon-intron circRNAs (EIciRNAs) [12], and intergenic circRNAs (Figure 1) [13]. circRNAs exert their functions mainly in the following ways: (1) circRNAs can act as competitive endogenous RNA or miRNA sponges: Studies have shown that the regulation of target genes by miRNAs is controlled by circRNAs [14]. circRNAs are rich in a large number of miRNA-binding sites, which compete with target messenger RNA (mRNA), and such circRNAs are called competitive endogenous RNAs (ceRNAs), also known as miRNA sponges [9]. Specifically, miRNAs can prevent the translation of target mRNAs by binding to the miRNA response element (MRE) in the 3’-UTR of the target mRNA, which affects the stability of the target mRNA and regulates its expression in the nucleus by binding to the promoter [15,16,17]. However, many circRNAs do not appear to bind miRNAs extensively because most of them in mammals are expressed at low levels and they rarely contain multiple binding sites for the same miRNA [18]. Nonetheless, many recent studies have found that circRNAs are a relatively common miRNA sponge, especially in cancer [19]. (2) circRNAs can act through related proteins: circRNAs can interact with different proteins, such as RNA-binding proteins (RBPs) or ribonucleoprotein complexes (RNPs), to form specific circRBPs or circRNPs, which in turn affect the mode of action of related proteins [20,21]. (3) circRNAs can regulate transcription: Although most circRNAs are localized in the cytoplasm, circRNAs generated from processed intron lariats (ciRNA) or from reverse splicing from retained introns (EIciRNA) are restricted to the nucleus of human cells [20]. Some nuclear-retained circRNAs are implicated in transcriptional regulation. For example, both exon-intron circRNAs circ-EIF3J and circ-PAIP2 can bind to U1 snRNP and further interact with RNA Pol II, thereby enhancing the expression of their parental genes in HeLa and HEK293 cells [12]. (4) circRNAs can be translated: For a long time, most researchers believed that circRNAs were a unique class of endogenous non-coding RNAs. Two studies by Legnini et al. and Pamudurti et al. broke down this “bias”. Legnini et al. found that circZNF609 (derived from the zinc finger protein 609 gene) can be translated into protein in a splicing-dependent and cap-independent manner in mouse and human skeletal muscle. However, the translational activity of circular templates is much lower compared with linear RNAs [22]. Pamudurti et al. found a circRNA on the Drosophila head, circMbl, which can translate proteins through cap-independent translation [23]. This discovery endows circRNAs with new possibilities and provides a new direction for future research on circRNAs. However, this also raises new questions about the concept of circRNAs, which are generally regarded as non-coding RNAs. (5) There are some other functions of circRNAs, such as regulating the stability of mRNAs [24], transporting from the cell body to the extracellular fluid through exosomes [25], possibly having the function of signaling molecules, etc. (Figure 2). It is believed that functions will be discovered in the future.

Radiotherapy, the first-line treatment for more than half of cancer patients, applies ionizing radiation to target and kill tumors. It is especially important to improve the local control rate and long-term survival quality of malignant tumors. However, a small portion of patients tolerate or even resist radiotherapy, which eventually leads to tumor recurrence or distant metastasis [26]. Therefore, clarifying the mechanisms of radiosensitivity and exploring specific biomarkers of radioresistance are hotpots in antitumor therapy. In addition, radiation injury caused by radiation doses that exceed the tolerance of tissues or organs is also a major challenge that limits the efficacy of radiotherapy. So, how to predict radiotherapy injury early? How to mitigate radiotherapy injury? These are questions to ponder. By far, quite a few studies have found that circRNA is closely associated with tumor cell radioresistance and injury. Radiation exposure induces significant changes in circRNA expression levels in cancer cells. circRNA competitively binds to miRNAs through ceRNA crosstalk, thus affecting the expression of genes related to radioresistance downstream of miRNAs, and ultimately leading to corresponding changes in the DNA damage repair pathway, epithelial–mesenchymal transition process, Wnt/β-catenin pathway, and tumor stem cells. In this review, we mainly focus on the expression changes and regulatory mechanisms of circRNAs related to radioresistance and radiation injury and prospect the clinical translation of circRNA in the field of tumor radiotherapy and its challenges.

CircRNAs have been shown to play important roles in tumor growth, metastasis, and therapy resistance [27]. There have been many reviews and discussions on the role of circRNAs in tumor growth, metastasis, and drug resistance, but there are few discussions on the use of circRNAs in radiotherapy (RT). Therefore, this paper discusses the recent research on circRNAs and RT, and hopes to provide new ideas on how circRNAs can improve the sensitivity of RT through a general discussion and summary of this field (Table 1) (Figure 3 and Figure 4).

## 2. CircRNAs in Tumor Radiosensitivity

Glioma: Glioblastoma (GBM) is the most common primary malignant brain tumor, and despite aggressive multimodal therapy everywhere, survival remains low. The current standard of care includes maximal surgical resection in combination with RT and temozolomide (TMZ) [60]. RT, as one of the main control methods for gliomas, has always been affected by its susceptibility to radiation tolerance. Recent studies have found that the expression of circRNAs seems to be able to regulate the radiosensitivity of glioma cells. For example, circ-000834 can lead to the up-regulation of RNF2 by the sponge miR-433-3p, which in turn causes glioma cells to develop radioresistance. Therefore, Di et al. believed that by knocking down circ-0008344, the miR-433-3p/RNF2 signaling axis can be further regulated, thereby enhancing radiosensitivity in gliomas [28]. circ-VCAN can directly bind and negatively regulate miR-1183 by acting as a molecular sponge, thereby playing a role in the radioresistance of glioma. Overexpression of circ-VCAN can accelerate the proliferation, migration, and invasion of glioma cells after irradiation, and inhibit apoptosis [29]. Guan et al. found that circPITX1 overexpression can promote the glycolysis process and make gliomas radioresistant, but 2-DG (a glycolysis inhibitor) can counteract the effect of promotion. Down-regulation of circPITX1 inhibits glycolysis, thereby sensitizing glioma cells to radiation. Further study found that it works by regulating the miR-329-3p/NEK2 axis through molecular sponge action [30]. Other researchers believe that there is also a relationship between radiation and circRNAs in exosomes. Wang et al. found that the content of circRNAs in exosomes increases after low-dose irradiation, but high-dose radiation does not significantly promote the secretion of exosome circRNAs. Further study found that low-dose radiation-induced circ-METRN in exosomes can promote glioma progression and radiation resistance, which may play a role through the miR-4709-3p/GRB14/PDGFRα pathway [31]. In addition, some studies have suggested that the expression of circRNAs in extracellular vesicles (EVs) secreted by glioma cells may also be related to their radiation resistance. By analyzing the differential circRNAs expression profiles between EVs isolated from U251 cells and EVs isolated from radiation-resistant U251 (RR-U251) cells and using bioinformatics, they found that circATP8B4 in RR-EVs can act as a miR-766 sponge, which may be associated with glioma radioresistance [32]. These studies have shown that circRNAs can affect the radiosensitivity of gliomas through a variety of pathways. However, most of these studies are limited to the cellular level and have not been validated in animal models, and the difficulty in constructing animal models of glioma may be one of the reasons. In addition, thanks to the development and extensive use of the circRNA database, researchers can more easily obtain relevant data, and it is expected that the research results of circRNAs in glioma can be applied to the clinic more quickly.

### 2.1. Head and Neck Tumors

Nasopharyngeal carcinoma (NPC): RT has always been a main method for the treatment of NPC. With advances in radiotherapy technology, the prognosis of most patients with NPC has improved greatly, but there are still some patients with poor radiotherapy sensitivity. circRNAs also seem to be closely related to the efficacy of RT in NPC. Chen et al. pointed out that circRNA-000543 can regulate the radiosensitivity of NPC. They found that circRNA-000543 is up-regulated in radioresistant NPC patients, and circRNA-000543 can act as a sponge for miR-9. Silencing circRNA-000543 could sensitize NPC cells to irradiation by targeting the miR-9/PDGFRB axis [33]. Through bioinformatics analysis, Yang et al. predicted that curcumin could interact with multiple potential circRNA–miRNA-mRNA pathways, thus enabling sensitization to radiotherapy. They further verified experimentally that hsa-circRNA-00060 expression levels were significantly down-regulated in curcumin-treated irradiated cells, and it could act by regulating the downstream target gene epidermal growth factor receptor (EGFR) by sponging miR-1276 [34]. Interestingly, another related study showed that curcumin can restore the radiosensitivity of NPC cells by regulating the expression changes of the hsa-circRNA-102115/miR-335-3p/MAPK1 interaction network [61]. Zhu et al. studied the circulating serum exosomes of 210 patients with NPC and found that circMYC is differentially expressed in the serum exosomes of radiosensitive and radioresistant patients, and circMYC in the serum of radioresistant patients is significantly higher in exosomes than in radiosensitive patients. Therefore, they considered circulating exosomal circMYC as a biomarker and potential therapeutic target for NPC [62]. circRNA-000285 is also significantly up-regulated in NPC tissues and serum. The level of circRNA-000285 in radioresistant NPC patients is significantly increased by three times compared with radiosensitive NPC patients, suggesting that circRNA-000285 may serve as a novel biomarker for NPC and is associated with the radiosensitivity of NPC [63]. Likewise, hsa-circRNA-001387 is differentially expressed in radiosensitive and resistant patients with NPC. One study found that the expression level of hsa-circRNA-001387 in radiosensitive NPC patients was significantly lower than that in radioresistant patients. In addition, the overall survival and progression-free survival were shorter in patients with higher expression of hsa-circRNA-001387. These findings make hsa-circRNA-001387 a promising biomarker for predicting the efficacy of radiotherapy for NPC [64]. It is hoped that circRNAs can be used as an auxiliary diagnostic method in the clinic as soon as possible, and more research on the mechanism of circRNAs in the regulation of radiosensitivity of NPC will appear in the future.

Other head and neck tumors: The regulation of radiosensitivity in other head and neck tumors is also thought to be related to circRNAs. Chen et al. found that up-regulation of circATRNL1 can increase the radiosensitivity of oropharyngeal squamous cell carcinoma by sponging miR-23a-3p, reducing colony formation and cell growth, and inducing apoptosis and cell cycle arrest, and is also able to block endogenous inhibition of the target gene PTEN. Moreover, they also examined the expression level of the Akt signaling pathway after circATRNL1 overexpression, confirming that it is implicated in this process. However, whether circATRNL1 affects the radiosensitivity of OSCC by mediating Akt signaling after irradiation is unclear and requires further study [35]. Wu et al. found that the radioresistance of hypopharyngeal cancer cells promoted by circux1 is dependent on the caspase 1 pathway, and knockout of circux1 can significantly improve the radiosensitivity of hypopharyngeal cancer cells. The stable expression of circux1 in the cytoplasm depends on m6A modification mediated by mettl3 [36]. Therefore, there is a conjecture that can the stability of circRNAs can be regulated by changing m6A modification, thereby affecting the expression of these circRNAs that act as the “culprit” for radioresistance.

### 2.2. Gastrointestinal Cancer

Esophageal cancer (EC): EC is the eighth most common cancer in the world and the sixth most common cause of cancer-related death, with a relatively low five-year overall survival rate. In addition to the known histopathological and epidemiological differences, the molecular features of EC can be differentiated between esophageal squamous cell carcinoma and esophageal adenocarcinoma [65,66]. Esophageal squamous cell carcinoma is more similar to squamous cell carcinoma of other organs than esophageal adenocarcinoma [67]. In China, more than 95% of EC patients are diagnosed with esophageal squamous cell carcinoma [68]. RT is an indispensable part of the treatment of esophageal squamous cell carcinoma, especially in locally advanced esophageal squamous cell carcinoma [69]. However, radioresistance has always been one of the reasons for imprisoning the efficacy of RT for EC [70]. Liu et al. found that circRNA-100367 is highly expressed in human radioresistant esophageal squamous cell carcinoma cell lines. Silencing circRNA-100367 inhibits the proliferation and migration of esophageal squamous cell carcinoma cells. They further found that circRNA-100367 cooperates with miRNA-217 to regulate the radioresistance of esophageal squamous cell carcinoma cells through Wnt3 [37]. Ma et al. observed that circPRKCI and PARP9 are up-regulated in esophageal squamous cell carcinoma tissues and cells, whereas miR-186-5p is down-regulated. In addition, down-regulation of circPRKCI reduces the tumor growth of EC cells, inhibits cell viability, colony formation, and cell cycle progression, and increases the radiosensitivity of EC cells. Furthermore, it was found that circPRKCI is a sponge of miR-186-5p, and down-regulation of circPRKCI can inhibit EC progression and improve cell radiosensitivity by regulating the miR-186-5p/PARP9 axis. Therefore, it is believed that circPRKCI may improve the prognosis of EC patients [38]. circVRK1 is decreased in ESCC tissues, and patients with low circVRK1 levels have a poor prognosis. The possible reason is that circVRK1 can act as a sponge for miR-624-3p to inhibit EMT by reducing PTEN/PI3K-mediated AKT activity, thereby enhancing the sensitivity of ESCC cells to RT [39]. In addition, multiple circRNA–miRNA regulatory networks have been found. For example, circ-0014879 can also lead to the radioresistance of ESCC by regulating the miR-519-3p/CDC25A pathway [40]. Silencing circ-0000554 also inhibits EC progression and enhances cellular radiosensitivity by downregulating FERMT1 by sponging miR-485-5p [41]. Su et al. used the Arraystar CircRNA microarray technique, hoping to explore the expression pattern of circRNA between the EC radiation-resistant cell line KYSE-150R and its parental cell line KYSE-150, so as to explore the mechanisms of acquired radiation tolerance of EC. They further observed and validated the differential expression profile of circRNAs in radiation-resistant EC cells and compared them with parental EC cells. Two potential key circRNAs were found, namely circRNA-001059 and circRNA-000167, and the pathway analysis suggested that the Wnt signaling pathway may also be implicated in radiation resistance [71]. However, the specific mechanism may need further exploration.

Gastrointestinal tumors: The Intergroup 0116 (INT-0116) experiment confirmed the significant role of adjuvant chemoRT in the radical operation of gastric cancer, which established the important position of RT in the treatment of gastric cancer [72]. Recently, studies have found that circRNAs may be implicated in the regulation of radiosensitivity in gastric cancer. Shao et al. found that down-regulation of circ-DONSON can enhance the radiosensitivity of gastric cancer cells, which accelerates GC progression through the miR-149-5p/LDHA axis, inhibits cell proliferation, migration, invasion, and angiogenesis to a certain extent, and enhances cell apoptosis, suggesting that it may be a promising biomarker for GC therapy [42]. This may provide new ideas for exploring the improvement of the radiosensitivity of gastric cancer. circRNAs are also implicated in the regulation of the radiosensitivity of colorectal cancer. Gao et al. detected colon cancer tissues and cells and found that the expressions of circ-0055625 and MSI1 were significantly increased compared with normal tissues and cells, and RT can also increase the expressions of circ-0055625 and MSI1 in colon cancer cells. Further studies showed that circ-0055625 can act as a sponge for miR-338-3p, and knockout of circ-0055625 can block cell proliferation, migration, and invasion by regulating the miR-338-3p/MSI1 axis, and improve colon cancer cell proliferation, apoptosis, and radiosensitivity. This result may provide a rationale for improving the treatment of colon cancer RT. However, the authors did not explore the effect of the miR-338-3p-regulated colon cancer process and radiosensitivity, which may need to be explored in further studies [43]. In addition, circRNAs in exosomes also have regulatory effects on the radiosensitivity of colorectal cancer. Studies have shown that knockout of circ-IFT80 in exosomes can reduce the radiosensitivity and apoptosis of colon cancer cells by regulating the miR-296-5p/MSI1 axis, promoting CRC cell proliferation and accelerating the cell cycle [44]. Another study showed that RT led to a significant increase in the expression of circ-0067835 in serum exosomes, and knockout of exosome-mediated circ-0067835 could inhibit cell proliferation and cell cycle progression and enhance cell apoptosis and radiosensitivity. Then, it found that circ-0067835 knockout can regulate GF1R expression by sponging miR-1236-3p, thereby enhancing the radiosensitivity of CRC cells and inhibiting CRC progression [45]. Wang et al. found that circ-0001313 is significantly up-regulated in radiation-resistant colon cancer tissues. In addition, colon cancer tissues and cell lines under irradiation show high expression of circ-0001313 and low expression of miR-338-3p. Further study found that circ-0001313 acts as a sponge for miR-338-3p in colon cancer cells. Down-regulation of miR-338-3p can reverse the effects of the down-regulation of circ-0001313 on colon cancer cell viability, colony formation, and caspase-3 activity. Therefore, it is believed that knocking out the circ-0001313 gene can induce the radiosensitivity of colon cancer cells by negatively regulating miR-338-3p [46].

Hepatopancreatic tumors: cZNF292 can be induced in hepatoma cells in a time-dependent manner under hypoxia. This alteration is independent of hypoxia-inducible factor (HIF)-1α. Knockdown of cZNF292 increases SRY (sex determination region Y)-box9 (SOX9) nuclear translocation and subsequently reduces WNT/β-catenin pathway activity, resulting in hypoxic hepatocellular carcinoma cell proliferation rates, angiogenic mimicry, and radioresistance suppression. This suggests that cZNF292 can be used as a target to improve the radiosensitivity of hypoxic hepatocellular carcinoma cells [47]. Chen et al. identified and annotated 12,572 circRNAs in irradiated human pancreatic cancer cell exosomes by using RNA-seq analysis, of which 3580 circRNAs were annotated as known circRNAs. Through bioinformatics prediction, they found that hsa-circ-0002130 can bind hsa-miR-4482-3p and target NBN, and patients with high NBN expression have a poorer survival rate. However, they did not conduct further experiments to confirm their regulatory relationship and other functions of circRNAs. In addition, the sample size for RNA sequencing is smaller, and patient samples are more difficult to collect. Therefore, the effect on patients remains to be further verified [48]. In conclusion, this study provides new ideas for exploring the mechanism and treatment of circRNAs in pancreatic cancer.

Lung cancer: Abnormal circRNAs commonly exist in the occurrence and development of lung cancer, and play an oncogenic or inhibitory role, which can affect cell function. They can regulate cell proliferation, migration, and apoptosis through different signaling pathways, induce multidrug resistance, and modulate the tumor microenvironment (TME) and immune evasion [73]. By mining public data, Fan et al. constructed a circRNAs–miRNAs–ArmRNAs network. In addition, they also found that these armRNAs are contained in three autophagy-related genes (ARGs), namely regulation of autophagy, macroautophagy, and chaperone-mediated autophagy. By analyzing the correlation between ARGs and RT response in NSCLC patients, they found the clinical significance of autophagy in RT response in NSCLC patients, and the overall survival rate of RT patients with higher ARGS scores is significantly shortened. Therefore, it is believed that the regulatory mechanism of NSCLC RT can be predicted through this circRNAs–miRNAs–ARmRNAs–ARGS network [74]. Huang et al. found that circPVT1 is a sponge of miR-1208, and inhibiting circPVT1 can up-regulate miR-1208 to block the PI3K/AKT/mTOR signaling pathway, thereby improving the radiosensitivity of NSCLC cells [49]. Jin et al. found that down-regulation of circ-0086720 can enhance the sensitivity of NSCLC to RT by modulating the miR-375/SPIN1 axis, enhancing the improvement of RT efficacy in NSCLC [50]. AEG-1 has been reported to be implicated in the occurrence and progression of NSCLC [75,76], and further experiments by Li et al. verified that AEG-1 can promote NSCLC cell proliferation, migration, and invasion and develop radioresistance and chemoresistance. Subsequent studies found that the circRNA Circmtdh.4 is also implicated in this process and regulates the chemoresistance and radioresistance of NSCLC cells through the Circmtdh.4/miR-630/AEG-1 axis [51]. CircZNF208 is also found to be associated with radiosensitivity in NSCLC, and up-regulation of CircZNF208 results in the radioresistance of tumor cells. More interestingly, CircZNF208 can regulate the radiosensitivity of NSCLC cells via the miR-7-5p/SNCA axis under X-ray irradiation rather than under carbon ion irradiation. Therefore, Liu et al. believed that CircZNF208 can be used as a biomarker to select radiosensitive NSCLC patients for X-ray RT or CIRT, and guide the selection of the appropriate RT modality in the clinic [52]. Their team subsequently found that there are also some circRNAs that can also be used as biomarkers for the diagnosis and prognosis of NSCLC in CIRT or conventional radiation (X-ray or γ-ray) RT. First, they examined the expression of circRNAs in different radiosensitive NSCLC cell lines, including acquired and intrinsic radioresistance, rather than comparing expression changes before and after irradiation. A large number of differentially expressed circRNAs were then identified in cell lines from different genomic locations. circRNA sequencing data showed that compared with parental cells, 40 circRNAs are significantly up-regulated and 184 circRNAs are down-regulated in radiation-resistant cells. In addition, the radioresistant A549-R11 cell line has another feature similar to its sensitivity to carbon ions compared to its parental A549 cell line. This suggests that circRNAs with differential expression between the two cell lines may be potential biomarkers to differentiate radiation responses after X-ray or carbon ion irradiation. Finally, by combining high-throughput data screening and bioinformatics, an ncRNA network was constructed to describe the possible regulatory mechanisms of NSCLC in low LET-X-ray and high-LET carbon ion sensitivity [77]. Zhang et al. found that circRNAs can also act as tumor suppressor genes, and Circ-0001287 can inhibit the proliferation and metastasis of NSCLC cells and enhance their radiosensitivity. The specific mechanism is that Circ-0001287 can sponge miR-21 to up-regulate the expression of PTEN, thereby inhibiting the proliferation, metastasis, and radiation resistance of NSCLC cells [71]. These studies basically focus on the category of non-small cell lung cancer and fail to address the difficult issue of small cell lung cancer. Part of the reason may be that RT is not the preferred treatment for small cell lung cancer in the current treatment methods.

### 2.3. Reproductive System Tumors

Prostate tumors: circ-CCNB2 may predict the development of radioresistance in prostate cancer RT [54]. Cai et al. demonstrated through in vivo experiments that down-regulation of circ-CCNB2 can restore the radiosensitivity of prostate cancer through miR-30b-5p/KIF18A (kinase family member 18A)-mediated autophagy injury. At the therapeutic level, inhibition of circ-CCNB2 may have a radiosensitizing effect to improve RT efficacy in patients with recurrent prostate cancer [54]. TR4 has been confirmed to be associated with the development of prostate cancer and the development of chemoresistance [78,79]. On the one hand, TR4 can act as a tumor suppressor in prostate cancer [78], and on the other hand, it enhances the chemoresistance of prostate cancer [79]. Chen et al. found that TR4 can also affect the radiosensitivity of prostate cancer by regulating circRNAs [55]. First, radiation can increase the expression of TR4, and TR4 can reduce the radiosensitivity of prostate cancer cells by up-regulating the expression of circZEB1. Then they found that up-regulating circZEB1 can increase the expression of ZEB1 by sponging miR-141-3p, thereby making the prostate cancer cells develop radioresistance. Taken together, they concluded that radiation-induced TR4 expression affects the radiosensitivity of prostate cancer through the QKI/circZEB1/miR-141-3p/ZEB1 axis, making prostate cancer radioresistant, and targeting TR4 therapy may be a solution [55]. Up-regulation of circ-ZNF609 is found to increase the radioresistance of prostate cancer cells by sponging miR-501-3p leading to HK2 up-regulation and enhancing glycolysis. In addition to this, circ-ZNF609 also enhances the viability, migration, and invasion of prostate cancer cells through the miR-501-3p/HK2 axis, while hindering the apoptosis of prostate cancer cells [56]. circ-0062020 is also found to be up-regulated in prostate cancer tissues and cells. Further studies found that circ-0062020 can also regulate the miR-615-5p/TRIP13 axis by acting as a sponge, resulting in radioresistance of prostate cancer cells [57].

Uterine tumors: Gu et al. demonstrated that hsa-circ-0001610 has a progressive effect on the radioresistance of endometrial cancer cells, and clarified that the high expression of hsa-circ-0001610 in tissues may depend on tumor-associated macrophages (TAM)-derived exosome transport. TAM-derived exosomes can serve as a carrier for hsa-circ-0001610 to transfer hsa-circ-0001610 to endometrial cancer cells to sponge miR-139-5p and release cyclin B1 expression, thereby weakening the radiosensitivity of cancer cells. However, due to objective limitations, they are unable to collect enough radioresistant endometrial cancer patient samples to verify the clinical relevance of this mechanism. Despite some drawbacks, this study identifies a novel mechanism by which TAMs communicate with endometrial cancer cells and reduce their radiosensitivity, suggesting hsa-circ-0001610 as a potential intervention target to improve endometrial cancer radiosensitivity [58]. By X-raying cervical cancer HeLa cells with 10 Gy, Yu et al. constructed a circRNA–miRNA target gene interaction network to further study the regulatory role of circRNAs in radioresistance. This network suggests that circRNAs may play a central regulatory role. Therefore, circRNAs may play a major role in the response to radiation [80]. Zhao et al. demonstrated that hsa-circ-0009035 can target the miR-889-3p/HOXB7 axis as an important regulator of cervical cancer progression and radioresistance by exerting a competitive endogenous role, which can become a potential therapeutic target for cervical cancer therapy. However, there may be other miRNA/mRNA axes regulated by hsa-circ-0009035 yet to be revealed, which need to be discovered by further research [59].

Hematologic tumors: In the clinical treatment of hematological malignancies, allogeneic bone marrow transplantation can be performed, while whole body irradiation can be used to induce immunosuppression to eradicate malignant cells and prevent rejection of the donor bone marrow [81]. Wang et al. demonstrated through in vitro experiments on mouse BMMSCs that circRNA-014511 can inhibit the expression of P53 through the endogenous competitive binding of mmu-miR-29b-2-5p and induce a variety of apoptosis-related proteins and cell cycle-related proteins, reducing the radiosensitivity of cells. This may help reduce the loss of BMMSCs in the bone marrow of hematopoietic stem cell transplant patients undergoing total body irradiation and may play an active role in hematopoietic reconstitution and reduce the occurrence of graft-versus-host disease after transplantation [82]. Silencing circRNA-016901 can also improve the radioresistance of cells by modulating the miR-1249-5p/HIPK2 axis, thereby attenuating radiation-induced damage to bone mesenchymal stem cells [83].

## 3. CircRNA and Radiation Injury

As one of the main treatments for malignant tumors, radiotherapy plays a major role in the battle against malignant tumors. RT uses ionizing radiation to target and kill tumors. Ionizing radiation can damage DNA directly or indirectly through the generation of intermediate ions and free radicals, and trigger a complex and highly regulated DNA damage response that cascades multiple signaling pathways to initiate repair pathways that affect the radiosensitivity of cells. However, when the radiation dose exceeds the tolerance value of surrounding tissues or organs, it will lead to irreversible radiation damage. Although precision radiotherapy is highly developed, damage to surrounding normal tissues or organs from radiation exposure is still unavoidable, which largely affects patient outcomes and quality of survival. Recently, some researchers have found that ionizing radiation can induce alterations in circular RNA expression levels. Additionally, changes in circular RNAs modulate ionizing radiation response by targeting key signaling pathways, including DNA damage and repair, apoptosis, glycolysis, cell cycle arrest, and autophagy by acting as miRNA sponges and transcriptional regulators and binding to proteins in a variety of ways [84,85,86]. For example, circRSF1can act as a miR-146a-5p sponge to increase the expression of Ras-associated C3 botulinum toxin substrate 1 (RAC1), thereby enhancing the inflammatory and fibrotic phenotype of hepatic stellate cells, suggesting that circRSF1 may act as a preventive and a potential target for the treatment of radiation-induced liver injury [87]. Chen et al. used the chip technique to find that the circRNA/miRNA regulatory pathway is implicated in the process of radiation-induced liver fibrosis. Inhibition of hsa-circ-0071410 increased the expression of miR-9-5p, resulting in attenuated irradiation-induced activation of hepatic stellate cells. This study revealed the expression profiles and potential functions of differentially expressed circRNAs after irradiation. It provides new clues for the study of radiation-induced liver fibrosis [88]. Luo et al. identified 27 differentially expressed miRNAs by analyzing the transcriptomic profile of rat esophagus injury after ionizing radiation, among which 7 were down-regulated and the rest were up-regulated. A total of 197 differentially expressed circRNAs were identified, of which 110 were down-regulated and 87 were up-regulated. The functions of differentially expressed miRNAs and circRNAs were subsequently analyzed. These differentially expressed miRNAs are implicated in many cellular processes, such as cell proliferation, cell migration, and lipid metabolism. Among these processes, the most important one is lipid metabolism, especially steroid metabolism. The expression of sphingolipid metabolism-related circRNAs changed significantly during radiation, suggesting that sphingolipid metabolism may be an important link in radiation-induced esophageal injury. However, not all sphingolipid disorders are associated with the appearance of abnormal miRNAs/circRNAs [89]. To study the link between circRNAs and miRNAs and radiation-induced lung injury, Li et al. used a mouse model of radiation-induced lung injury to study circRNAs and miRNAs. By comparing the transcriptome profiles before and after irradiation, they found 21 significantly up-regulated and 33 significantly down-regulated miRNAs, and 17 up-regulated and 10 down-regulated circRNAs. GO analysis was then performed on the miRNAs of these apparently differentially expressed circRNAs. These differentially expressed miRNAs were found to be implicated in a series of cellular processes, such as positive regulation of α-T cell proliferation, the interstitial matrix, collagen fiber organization, chemokine receptor activity, cellular defense response, the B-cell receptor signaling pathway, etc. The important role of immune-related molecular pathways in early response after RT was revealed. Therefore, it is believed that, in the future, studying the mechanism of circRNAs such as circRNA5229, circRNA544, and circRNA3340 and related miRNAs and conducting early intervention will be helpful for the treatment of radiation-induced lung injury [90]. circFOXO3 can protect cardiomyocytes from radiation-induced cardiotoxicity by reducing DNA damage and apoptosis. Mechanistically, the expression of circFOXO3 can reduce the expression levels of Bax, caspase 3, and caspase 7 and increase the expression level of Bcl-2 [91]. Yang et al. first used the C57BL/6 mouse radiation-induced intestinal toxicity model to further study the protective effect of radiation on the gastrointestinal tract. To elucidate the underlying mechanism of the FIH/HIF axis in ionizing radiation-induced gastrointestinal syndrome, whole transcriptome analysis was performed in the control and N-oxalyl-D-phenylalanine (NOFD) pretreatment groups, and their ceRNA regulatory networks were obtained. After the lncRNA/circRNA-related ceRNA network was constructed, four key hub genes, namely EGFR, HIF1A, NOS2, and CDKN1A, were identified in this complex RNA crosstalk. Two new circRNAs (circRNA-2909 and circRNA-0323) and two lncRNAs (NONMMUT140549.1 and NONMMUT148249.1) were identified as important upstream regulators. Among them, these four key RNAs can promote the expression of HIF1A and NOS2 in the HIF-1 pathway by adsorbing microRNAs, especially mmu-miR-92a-1-5p. Combining all these results, they proposed that HIF inhibitors such as NOFD may be a therapeutic strategy for gastrointestinal radioprotection [92]. With the above findings, we found that the level of circRNA expression in cells is significantly altered after radiation exposure. This indirectly suggests that circRNAs are closely related to the radiation response. In addition, given the highly conserved, tissue-specific, and structurally stable properties of circRNA, an attempt could be made to develop circRNA as a biomarker for monitoring radiation damage.

## 4. Discussion

circRNAs, a novel class of endogenous non-coding RNAs with 3′ and 5′ ends covalently joined to form circular loop structures without free ends, are insusceptible to RNase R and exonucleolytic degradation. Additionally, they exert biological functions by acting as a molecular sponge for miRNAs, regulating gene transcription, participating in protein translation, and interacting with the binding protein RBP. Radiotherapy, which occupies a significant position in the whole management of tumors, has brought dawn for cancer patients. However, radioresistance and radiation injury are roadblocks to improving the efficacy of radiotherapy. Accumulating evidence has shown that circRNAs are widely involved in tumor radioresistance and radiation injury. Here, we reviewed the recent progress of circRNA in regulating radiosensitivity and radiation damage. This will provide a fresh perspective on the clinical translation of circRNAs in the field of anticancer therapy. Therefore, it also suggests that future research on circRNAs may be intensively investigated in several aspects. First of all, according to the ceRNA hypothesis, competitive binding can occur when circRNA and miRNA have binding sites, and circRNA plays a sponge role to influence the regulatory effect of miRNA on target genes. So, is it possible to develop certain circRNAs as a radiosensitizer by blocking the circRNA–miRNA regulatory pathway? Next, circRNA can be used as a template for translation to synthesize peptides or proteins. In the future, we can try to recombine the functional peptides encoded by circRNA with viral vectors or anti-drugs to construct anticancer vaccines or targeted drug delivery systems. In addition, as most cancers are insidious in origin and progress rapidly, they may have progressed to advanced stages at the time of diagnosis. Hence, the notion of “early detection, diagnosis, and treatment” remains the most powerful weapon in fighting against malignancy. It is urgent to explore early, fast and accurate biomarkers with superb specificity and sensitivity for diagnosis and prognosis. circRNA, which is currently a hot research topic, is an ideal choice for biomarkers related to screening, clinical staging, treatment efficacy, and prognostic risk stratification because of its stable structure, highly conserved sequence, and tissue expression specificity. In addition to the potential clinical applications mentioned above, there is one more point that deserves to be described for it and even explored in depth as a future research direction. A large quantity of studies has demonstrated that circRNA expression levels are rapidly altered in normal tissues after high doses of irradiation. This phenomenon would lead us to a series of thoughts: Can circRNA be used as a biomarker for predicting radiation damage? Can circRNA be developed as a radioprotective agent? At what doses of radiation does the expression of circRNAs change? If the irradiation dose is too high, does it lead to the degradation of circRNA? Do the functions of proteins in signaling pathways involved in DNA damage repair change with increasing exposure doses? The answers to these questions are still inconclusive and have yet to be resolved. Last but not least, although the rapid development of transcriptome sequencing technology and bioinformatics, the current status of circRNA research is at the summit of one’s power, many circRNAs have been annotated for their biological functions, there is still vast room for improvement. For example, a circRNA database that is applicable to multiple platforms, open source, and easy to maintain is yet to be constructed; unified and standardized naming principles of circRNA-encoded proteins are yet to be formulated; the effectiveness of circRNA extraction techniques and enrichment methods is yet to be broken through. Moreover, the existence of crosstalk cannot be ignored in conducting research on circRNA translation function, and it must be ensured that the detected circRNA translation products are derived from real circRNAs rather than peptides/proteins produced by crosstalk [23]. Of course, excluding crosstalk in these studies is quite tedious and will certainly add additional investment, so we expect to develop more convenient and economical assays. Above all, most of the current research on circRNAs is focused on basic research, and more series and comprehensive real-world cohort studies are needed to validate the feasibility of circRNAs for tumor biomarkers if clinical translation is to be achieved and patients are to benefit from them. Although the existing research results have done little to unravel the mystery of circRNAs, the research process of circRNAs will develop by leaps and bounds in the coming years through persistent efforts. We are one step closer to revealing the mechanism of circRNAs in radiosensitivity and achieving clinical translation. 

## 5. Conclusions

As a common treatment for tumors, RT has always had many limitations, such as radiation damage and radiation resistance. The high abundance and stability of circRNAs are undoubtedly promising targets for overcoming these limitations of RT. Therefore, we look forward to discovering more possibilities to solve these problems using circRNAs. Here, it was found that circRNAs are an important factor regulating radiosensitivity, especially in gastrointestinal tumors. On the one hand, most studies show that the expression of circRNAs can promote tumor growth and the formation of radioresistance, but on the other hand, some circRNAs have also been found to act as tumor suppressors and increase radiosensitivity. In addition, it was also found that by regulating the expression of circRNAs, the common clinical radiation damage problem can be solved. Of course, the current research on the relationship between circRNAs and RT is still scarce, too one-sided, and still in its infancy. Most studies are still focused on the relationship between circRNAs and radiosensitivity, which has not been practically used in clinical practice. However, it is believed that with the continuous development of genetic techniques and the continuous research on circRNAs, these difficulties and limitations will be overcome, and the research on circRNAs and RT will benefit more patients.

## Figures and Tables

**Figure 1 ijms-23-10444-f001:**
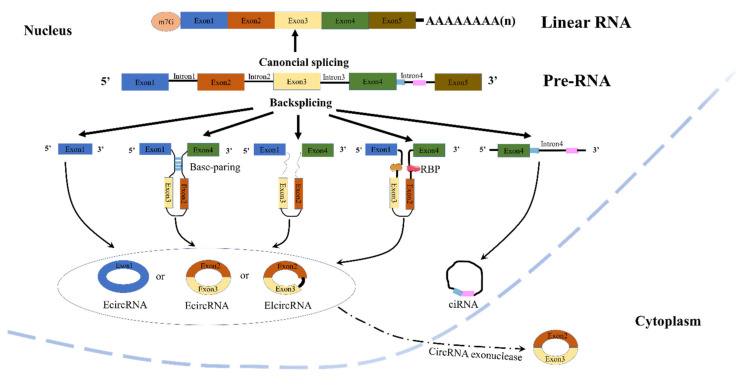
Biogenesis and classification of circRNAs.

**Figure 2 ijms-23-10444-f002:**
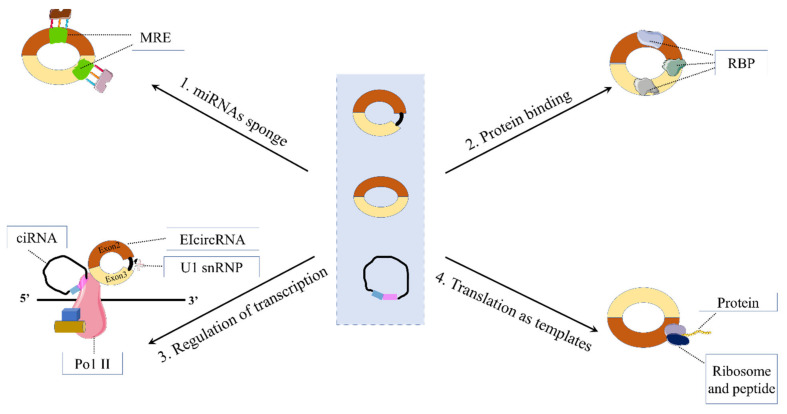
Functions of circRNAs.

**Figure 3 ijms-23-10444-f003:**
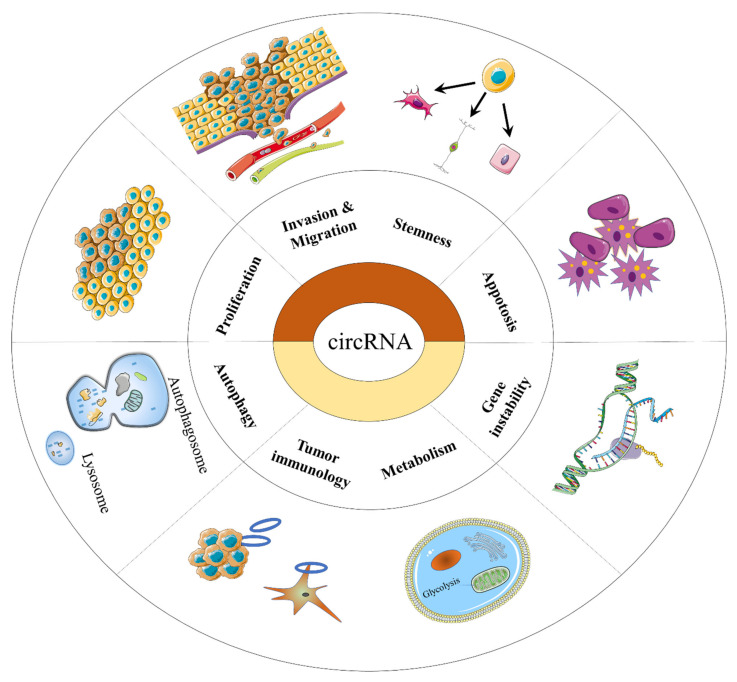
CircRNAs involved in the hallmarks of cancer.

**Figure 4 ijms-23-10444-f004:**
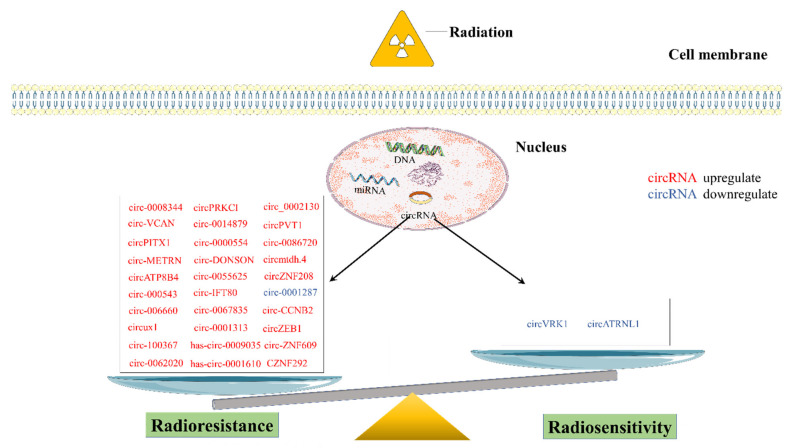
Role and regulatory pathways of cancer-related circRNAs.

**Table 1 ijms-23-10444-t001:** Mechanism and function of circular RNAs in regulating solid tumor radiosensitivity.

Cancer Type	CircRNA	Expression	Biological Function	Pathway	Refs
Glioma	circ-0008344	↑	Promotes radioresistance	circ-0008344/miR-433-3p/RNF2	[28]
	circ-VCAN	↑	Accelerates proliferation, migration, and invasion of glioma cells after irradiation, and inhibits apoptosis	circ-VCAN/miR-1183	[29]
	circPITX1	↑	Promotes radioresistance	circPITX1/miR-329-3p/NEK2	[30]
	circ-METRN	↑	Low-dose radiation-induced circ-METRN in exosomes promotes glioma progression and radioresistance	circ-METRN/miR-4709-3p/GRB14/PDGFRα pathway	[31]
	circATP8B4	↑	Promotes radioresistance through EVs	circATP8B4/miR-766	[32]
NPC	circ-000543	↑	Promotes radioresistance	circ-000543/miR-9/PDGFRB	[33]
	hsa-circ-006660	↑	Promotes radioresistance	circ-006660/miR-1276/EGFR	[34]
OSCC	circATRNL1	↓	Promotes radiosensitivity	circATRNL1/miR-23a-3p/PTEN	[35]
	circux1	↑	Promotes radioresistance	Caspase 1 pathway	[36]
EC	circ-100367	↑	Promotes proliferation and migration of ESCC cells and radioresistance	circ-100367/miR-217/Wnt3	[37]
	circPRKCI	↑	Promotes EC cells growth, cell viability, colony formation, cell cycle progression, and radioresistance	circPRKCI/miR-186-5p/PARP9	[38]
	circVRK1	↓	Promotes radiosensitivity	circVRK1/miR-624-3p/PTEN/PI3K/AKT pathway	[39]
	circ-0014879	↑	Promotes ESCC cells proliferation, migration and invasion, and radioresistance	circ-0014879/miR-519-3p/CDC25A pathway	[40]
	circ-0000554	↑	Promotes EC cells progression and radioresistance	circ-0000554/miR-485-5p/FERMT1	[41]
GC	circ-DONSON	↑	Promotes GC cells progression and radioresistance	circ-DONSON/miR-149-5p/LDHA	[42]
Colon cancer	circ-0055625	↑	Promotes tumor cells progression and represses apoptosis and radiosensitivity	circ-0055625/miR-338-3p/MSI1	[43]
	circ-IFT80	↑	Promotes tumor cells progression and radioresistance	circ-IFT80/miR-296-5p/MSI1	[44]
	circ-0067835	↑	Promotes tumor cells proliferation, cell cycle progression, radioresistance, and inhibits cell apoptosis	circ-0067835/miR-1236-3p/GF1R	[45]
	circ-0001313	↑	Promotes tumor cells viability, colony formation, and caspase-3 activity	circ-0001313/miR-338-3p	[46]
HCC	CZNF292	↑ (hypoxic)	Promotes tumor cells proliferation, angiogenic mimicry, and radioresistance	CZNF292/SOX9/WNT/β-catenin pathway	[47]
Pancreatic cancer	circ_0002130	↑	Promotes radioresistance	circ_0002130/hsa-miR-4482-3p/NBN	[48]
NSCLC	circPVT1	↑	Promotes radioresistance	circPVT1/miR-1208/PI3K/AKT/mTOR pathway	[49]
	circ-0086720	↑	Promotes radioresistance	circ-0086720/miR-375/SPIN1	[50]
	circmtdh.4	↑	Promotes NSCLC cell progression, and develops radioresistance and chemoresistance	circmtdh.4/miR-630/AEG-1	[51]
	circZNF208	↑ (X-ray)	Promote radioresistance	circZNF208/miR-7-5p/SNCA	[52]
	circ-0001287	↓	Inhibits NSCLC cells proliferation, metastasis, and radioresistance	circ-0001287/miR-21/PTEN	[53]
Prostate cancer	circ-CCNB2	↑	Promotes radioresistance	circ-CCNB2/miR-30b-5p/KIF18A	[54]
	circZEB1	↑	Promote radioresistance	circZEB1/miR-141-3/ZEB1	[55]
	circ-ZNF609	↑	Enhances the viability, migration, invasion, and glycolysis of PC cells, thus promoting radioresistance	circ-ZNF609/miR-501-3p/HK2	[56]
	circ-0062020	↑	Promotes radioresistance	circ-0062020/miR-615-5p/TRIP13	[57]
Endometrial cancer	hsa-circ-0001610	↑ (in TAM-derived exosomes)	Promotes radioresistance	hsa-circ-0001610/miR-139-5p/cyclin B1	[58]
Cervical cancer	hsa-circ-0009035	↑	Promotes tumor cells progression and radioresistance	hsa-circ-0009035/miR-889-3p/HOXB7	[59]

↑: over expression; ↓: low expression; NPC: nasopharyngeal carcinoma; OSCC: oral squamous cell carcinoma; EC: esophageal cancer; GC: gastric cancer; HCC: hepatocellular carcinoma; NSCLC: non-small cell lung cancer.

## Data Availability

Not applicable.

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
