# Peer review of "Mechanism and Function of Circular RNA in Regulating Solid Tumor Radiosensitivity"

_ijms, 2022, doi:10.3390/ijms231810444_

Round 1
Reviewer 1 Report
This is a very interesting and documented review that is written and structured in a precise and concise style.
However, this reviewer has only one general and nearly conceptual concern about the potential link between radiosensitivity and circular RNA : tumor response to radiotherapy, whether it is radiosensitive or radioresistant obeys a specific dose response. Hence, while the authors endeavor to establish a link between circular RNA and tumor response to radiation, there is nothing mentionned about a dose-dependent production (or disappearance ) of circular RNA and with what specificity for each tumor type. it should be discussed
In addition, the role of DNA damage repair in the response ot radiation is very well documented : if circular RNAs would regulate it, they should act very rapidly (less than hours after irradiation). To the contrary, the authors cite some studies about cellular and tissue events donwstream the DNA damage formation and repair , notably through inflammation process. How could microRNA influence DNA repair (by protein binding?) what is the rationale and the evidence? It should be discussed.
Finally, irradiation can break DNA but also RNA : maybe we can expect a RNA breakage and therefore a linearization of the circular RNA after irradiation as far as the cumulative dose increases. Therefore, the potential influence of circular RNA would decrease during the radiotherapy. The RNA breakage by irradiation should be discussed.
All the points can be discussed in the general discussion but they are very crucial to evaluate the relevance of the general hypothesis of a significant regulation of the response to radiation by circular RNA
Author Response
Replay: Thank you from the bottom of my heart for these constructive comments. Regarding your first question: We searched and systematically combed the relevant literatures about the specific irradiation doses involved in eliciting a tumor radiation response, and listed the results as follow. The commonly used irradiation dose range is 2-8 Gy in cell experiments. In animal experiments, it is approximately 6-10 Gy. However, there is no study that can definitively answer that a dose-dependent production (or disappearance) of circular RNA and with what specificity for each tumor type. But it is thought-provoking, we also delve into this issue in the later discussion section and consider it as a follow-up research direction for further exploration. For the second recommendation: radiotherapy is a cancer treatment that applies high doses of ionizing radiation to induce cell death, mainly by triggering DNA double-strand breaks. Differential expression of circular RNAs modulate ionizing radiation response by targeting key signaling pathways, including DNA damage and repair, apoptosis, glycolysis, cell cycle arrest, and autophagy by acting as miRNA sponges, and transcriptional regulators and binding to proteins RBP in a variety of ways. DNA damage response is one of the basic physiological mechanisms of tumor cells after radiation exposure. To be specific, circRNA regulates the DNA damage response repair pathway in post-radiation tumor cells in three main ways: (1) circRNA competitively binds to miRNAs, thereby affecting the expression of genes downstream of miRNAs related to the DNA damage repair pathways; (2) circRNA regulates the expression of key sensors and conductors in the DNA damage repair pathway at the transcriptional level; (3) circRNA affects cell cycle regulation. Most importantly, the ceRNA mechanism is a more studied regulatory model for circRNA. Based on your proposal, we have discussed it in the review. For one last suggestion, the lack of the 5′ end cap structure and the special structure of poly A at the 3′ end makes circRNA less susceptible to degradation by the nucleic acid exonuclease RNase R and therefore more stable than the related linear mRNA. Precisely because of its stable structure and highly conserved sequence, it is difficult to be degraded through conventional pathways. Ionizing radiation can induce alterations in circular RNA expression levels. However, there is no clear clue whether high dose of radiation induce circRNA changes through linearization of the circular RNA. This has aroused our great interest, and we will look into it in the discussion, so as to promote the new development of circRNA in radiotherapy.
In order to get a more professional editing, we have obtained English language editing help from MDPI Language Editing Services (https://www.mdpi.com/authors/english). All modified records are displayed in the manuscript for your review.
Reviewer 2 Report
Review of the manuscript ijms-1872435: Mechanism and Function of circular RNA in regulating solid 2 tumor radiosensitivity
The present review article on the role of circRNA in radiation therapy contains the essential aspects of the topic. It includes almost all tumor entities relevant to radiotherapy and the essential circRNA types relevant in these entities.
Although the language is good, there are still difficulties in understanding some paragraphs. Grammatical and semantic errors should be corrected.
1. Page 1, last paragraph: There are repetitive sentences in line 39/40 and 44/45.
2. Page 6, last paragrape, line 106-109: “Guan et al. found that circPITX1 overexpression can promote the glycolysis process 106 and make gliomas radioresistant, but 2-DG (glycolysis inhibitor) can counteract the effect 107 of promotion, indicating that circPITX1 promotes glycolysis by enhancing glycolysis, and 108 makes glioma cells radioresistant. The last half sentence does not fit here.
3. Page 7, lines 131-132: Sentence is not clear.
4. Page 7, lines 138-140: Role of curcumin is not clear from this statement. Does curcumin interact with a specific circRNA/miRNA/mRNA or overall network?
5. Page 7, lines 146-148: Sentence not clear, please edit.
6. Page 7, lines 154-157: Is Has-circRNA-001387 expressed higher or lower in the radiosensitive and resistant patients, make the statement more clear.
7. Page8, lines 175-176: Sentence should be edited.
8. Discussion is not well written. Rather, the importance of circRNA for radiotherapy and its improvement should be discussed.

Author Response
C1. Page 1, last paragraph: There are repetitive sentences in line 39/40 and 44/45.
R1. We are sorry about it. Now it has been corrected.
C2. Page 6, last paragrape, line 106-109: “Guan et al. found that circPITX1 overexpression can promote the glycolysis process 106 and make gliomas radioresistant, but 2-DG (glycolysis inhibitor) can counteract the effect 107 of promotion. The last half sentence does not fit here.
R2. Thanks. According to your suggestions, we have removed the last half sentence.
C3. Page 7, lines 131-132: Sentence is not clear.
R3. Thank you for your suggestion, we have rephrased this section.
C4. Page 7, lines 138-140: Role of curcumin is not clear from this statement. Does curcumin interact with a specific circRNA/miRNA/mRNA or overall network?
R4. Sorry. The lack of clarity in this statement has led you to such doubts. Now we have expressed this section clearly. Simply put, curcumin exerts a radiosensitizing effect. In this cited paper, the authors established a radio-resistant model with NPC cell line and treatd irradiated cell lines with curcumin, then analyzed the changes in circRNA expression levels before and after curcumin treatment to investigate the radiosensitization effects of curcumin. After that, bioinformatics analysis was used to predict multiple potential pathways for curcumin radiosensitization, and only hsa_circRNA_006660/ miR-1276/ EGFR pathway were verified by a series of experiments. The relevant content in the article has been changed to “Through bioinformatics analysis, Yang et al. predicted that curcumin could interact with multiple potential circRNA-miRNA-mRNA pathways, thus enabling sensitization to radiotherapy. They further verified experimentally that hsa_circRNA _00060 expres-sion levels were significantly down-regulated in curcumin-treated irradiated cells, and it could act by regulating the downstream target gene epidermal growth factor recep-tor (EGFR) by sponging miR-1276”.
C5. Page 7, lines 146-148: Sentence not clear, please edit.
R5. We are sorry about it. We have re-edited these sentences in our review.
C6. Page 7, lines 154-157: Is Has-circRNA-001387 expressed higher or lower in the radiosensitive and resistant patients, make the statement more clear.
R6. Thanks for your suggestion. We dug deeper into the study and found that the expression level of hsa_circRNA_001387 in radiosensitive NPC patients is significantly lower than that in radioresistant patients, and we have added this section to the review.
C7. Page8, lines 175-176: Sentence should be edited.
R7. We are sorry about it. We have re-edited the sentence in our review.
C8. Discussion is not well written. Rather, the importance of circRNA for radiotherapy and its improvement should be discussed.
R8. You are right. we realize that the need for improvement in this section. We have elaborated on the importance and applications of circRNA in the field of radiotherapy in more detail in that our discussion.
In order to get a more professional editing, we have obtained English language editing help from MDPI Language Editing Services (https://www.mdpi.com/authors/english). All modified records are displayed in the manuscript for your review.
Round 2
Reviewer 1 Report
The authors have reached my requirements satisfactorily.